# Profiling the Immune Response to Periprosthetic Joint Infection and Non-Infectious Arthroplasty Failure

**DOI:** 10.3390/antibiotics12020296

**Published:** 2023-02-01

**Authors:** Cody R. Fisher, Robin Patel

**Affiliations:** 1Mayo Clinic Graduate School of Biomedical Sciences, Department of Immunology, Mayo Clinic, Rochester, MN 55905, USA; 2Division of Clinical Microbiology, Department of Laboratory Medicine and Pathology, Mayo Clinic, Rochester, MN 55905, USA; 3Division of Public Health, Infectious Diseases, and Occupational Medicine, Department of Medicine, Mayo Clinic, Rochester, MN 55905, USA

**Keywords:** PJI, periprosthetic joint infection, arthroplasty, multi-omics, immune profiling

## Abstract

Arthroplasty failure is a major complication of joint replacement surgery. It can be caused by periprosthetic joint infection (PJI) or non-infectious etiologies, and often requires surgical intervention and (in select scenarios) resection and reimplantation of implanted devices. Fast and accurate diagnosis of PJI and non-infectious arthroplasty failure (NIAF) is critical to direct medical and surgical treatment; differentiation of PJI from NIAF may, however, be unclear in some cases. Traditional culture, nucleic acid amplification tests, metagenomic, and metatranscriptomic techniques for microbial detection have had success in differentiating the two entities, although microbiologically negative apparent PJI remains a challenge. Single host biomarkers or, alternatively, more advanced immune response profiling-based approaches may be applied to differentiate PJI from NIAF, overcoming limitations of microbial-based detection methods and possibly, especially with newer approaches, augmenting them. In this review, current approaches to arthroplasty failure diagnosis are briefly overviewed, followed by a review of host-based approaches for differentiation of PJI from NIAF, including exciting futuristic combinational multi-omics methodologies that may both detect pathogens and assess biological responses, illuminating causes of arthroplasty failure.

## 1. Total Joint Arthroplasty Failure

Total joint arthroplasty is a common restorative surgery. In the United States (US), more than 1.5 million people undergo total hip arthroplasty (THA) or total knee arthroplasty (TKA) yearly and it is anticipated that numbers of arthroplasties will rise due to the aging population, high rates of obesity, and physical activity throughout the lifespan, including in later years. It has been predicted that THA and TKA procedures will grow by 137% and 601%, respectively, in the US between 2005 and 2030, resulting in more than 5 million primary THAs and TKAs and an estimated US$1.85 billion in annual hospital-related costs by 2040 [1,2,3,4,5]. A study of over 1.5 million primary TKA or THA patients from the US National Inpatient Sample found the mean age of patients undergoing primary TKA to have decreased from 68 years in 2001 to 66 years in 2011, while the mean age of those undergoing primary THA decreased from 66 years in 2001 to 65 years in 2011. In total, 64% and 56% of those in the TKA and THA groups, respectively, were women [6].

There are several reasons patients undergo joint replacement surgery. Most commonly, the procedure is in response to symptomatic osteoarthritis, followed by inflammatory arthritides, such as rheumatoid arthritis, or joint damage due to injury, tumors, or osteoporosis; regardless of underlying disease, these conditions result in pain, loss of joint mobility, and/or an overall decrease in quality of life [7,8,9,10]. In most cases, joint replacement provides dramatic pain relief and restoration of joint function, although 1 to 3% of patients require revision surgery due to complications, such as periprosthetic joint infection (PJI) or non-infectious arthroplasty failure (NIAF) [11,12,13,14,15]. As primary arthroplasty procedures increase in numbers, so do revision surgeries. It is estimated that surgical revision procedures will grow from ~130,000 annually in 2014 to ~300,000 by 2030, with an increase from ~55,000 to ~85,000 for hip and ~72,000 to more than 200,000 for knee revisions. Not only are numbers of joint revisions increasing, but the rate of revisions in younger patients, particularly those between the ages of 55 and 64, is growing, with total increases of 9.1% and 8.6% for THA and TKA, respectively, from 2002 to 2014 [16]. A 2014 retrospective study of 120,538 patients with TKAs found that, one-year post surgery, those younger than 50 years old were more likely to develop arthroplasty failure than those 65 years or older. In that study, PJI developed in 1.4% of those under 50 years of age and 0.7% in those over 65 years of age, while NIAF developed in 3.5% and 0.8%, respectively [17]. Determining the underlying cause of arthroplasty failure, whether infectious or non-infectious, and in turn, choosing a suitable treatment approach is essential, albeit difficult in some cases [9].

### 1.1. Periprosthetic Joint Infection

PJI occurs in ~1 to 3% of patients undergoing primary total joint arthroplasty and makes up 20% to 50% of arthroplasty failures [9,13,16,18,19,20,21]. In 1995, the National Institutes of Health (NIH) Consensus Development Panel on Total Hip Replacement described PJI as a “devastating complication” that is “challenging” to diagnose [10]; antimicrobial agents plus surgery are needed to treat it. The type of surgical intervention is based on the infecting organism(s), timing and duration of infection, and clinical presentation. For acute PJI, joint irrigation, followed by debridement, long-term antimicrobials, and implant retention (DAIR), is typical, costing around US$67,000 [22]. For non-acute cases, resection of components is characteristically necessary, either as one- or two-stage procedures. Two-stage exchange arthroplasties typically cost around US$114,000, though costs vary [9,23,24,25,26]. In all, management of PJI costs hospitals ~5-fold more than uncomplicated hip arthroplasty, totaling approximately US$2 billion per year, not including non-medical costs [5,27,28]. This cost is in addition to the individual patient burden, often including devastating morbidity, expense, impairment of quality of life, and potential loss of implanted devices, or even limbs, in extreme circumstances [9,29]. There are several risk factors associated with PJI, including obesity, smoking, and immune-disrupting disorders and their treatments, such as diabetes mellitus and rheumatoid arthritis. Men have been observed to be more prone to infection than women, although the biological underpinning for this observation is unknown [30,31,32,33,34,35,36,37,38,39,40,41].

PJI is typically caused by the formation of bacterial biofilms on device surfaces and in the surrounding tissues; biofilm formation involves the production of extracellular polymeric substances and immune-modulating products that protect microorganisms from antimicrobial agents and the host immune response [42,43,44,45,46]. Staphylococci, primarily *Staphylococcus epidermidis* and *Staphylococcus aureus*, are the most common causes, followed by polymicrobial infections, streptococci, anerobic bacteria, aerobic Gram-negative bacilli, and enterococci. In rare cases, other bacterial types, or even fungi, cause PJI (Figure 1) [9,47,48,49,50,51,52]. Culture-negative infections make up 6 to 15% of PJI cases, although rates up to 42% have been reported. Culture-negative PJI is particularly challenging due to the difficulty in selecting a treatment regimen for an unknown causative organism (which may be “over” or “under” treated) or even in classifying the joint as infected in the first place [47,53,54,55].

### 1.2. Non-Infectious Arthroplasty Failure

NIAF includes causes of arthroplasty failure other than infection, accounts for ~50 to 80% of arthroplasty failures and can cost upwards of US$40,000 per TKA revision and US$15,000 per THA revision [16,56,57,58]. It is typically divided into aseptic loosening, periprosthetic fracture, instability, osteolysis/adverse tissue reaction, and other/miscellaneous subgroups, with classification dependent on clinician discretion in many cases. As with arthroplasty failure due to PJI, revision surgery is often used to treat NIAF (unlike PJI, without antimicrobial treatment) [9,11,12,14,15,59,60,61,62,63,64]. Studies conducted at the Mayo Clinic in 2017 and 2019 found that the distribution of causes of NIAF of TKAs included 36% aseptic loosening, 21% periprosthetic fracture, 21% instability, 19% osteolysis, and 2% arthrofibrosis, while the distribution of causes of THA revisions for NIAF included 37% periprosthetic fracture, 26% aseptic loosening, 19% adverse tissue reaction to the device, 13% instability, and 5% other, including implant failure/stem fracture, and iliopsoas tendinitis requiring repositioning (Figure 2) [12,64].

## 2. Current Arthroplasty Failure Diagnostic Techniques

Despite the health and financial impact of arthroplasty failure, there are no universally accepted clinical definitions or diagnostic criteria for PJI and NIAF. Between 2001 and 2021, there was an ~30-fold increase in PubMed yearly publications for all PJI and those specifically related to PJI (Figure 3). The influx of PJI-related content and rising PJI numbers have led to several organizations, such as the Infectious Diseases Society of America (IDSA) and the Musculoskeletal Infection Society (MSIS) in 2011, International Consensus on Orthopedic Infections Meetings in 2013 and 2018, and the European Bone and Joint Infection Society (EBJIS) in 2021, to propose diagnostic guidelines, although there is constant evolution and refinement as a result of new knowledge and improving diagnostic approaches; a global consensus definition of PJI has yet to be reached [25,65,66,67,68,69]. It has been recently argued that while complex, multi-step approaches to PJI diagnosis may be useful in research settings, a single accurate differentiative assay would be most helpful in clinical practice [70]. In addition to distinguishing PJI from NIAF, an important consideration is a need to define the microbial etiology of PJI in the infected cases. Whether used in combination or as individual analyses, current diagnostic assays are primarily either microbial-based, such as traditional bacterial culture and molecular analyses, or host-based, such as histopathological evaluation and measurement of single host-based biomarkers. While able to discern PJI from NIAF in some cases, these techniques come with limitations.

### 2.1. Microbial-Based Diagnostic Techniques

Bacterial culture and molecular assays are traditionally used as microbial-based techniques for PJI diagnosis and pathogen identification [71]. Traditional bacterial culture consists of harvesting patient samples, such as synovial fluid, sonicate fluid (i.e., fluid generated from sonication of resected implants), and/or periprosthetic tissue, and inoculating them into or onto culture media, which are then incubated to assess for microbial growth. If microbial growth is detected, that growth is identified by additional analysis, such as matrix-assisted laser desorption/ionization-time of flight (MALDI-TOF) mass spectrometry (MS) [71,72,73,74]. While bacterial culture remains the clinical gold standard for PJI diagnosis and allows recovery of microorganisms for antimicrobial susceptibility testing, it has limitations. Inherently, culture relies on growth in or on culture media such that inadequate growth conditions or low bacterial inocula can lead to negative results; further, culture is affected by antimicrobial usage before sampling (which is common). Due to these limitations, culture-based techniques may have imperfect sensitivity, even when infection is present. The type of sample analyzed also affects sensitivity. For example, periprosthetic tissue culture is generally less sensitive than sonicate fluid culture [73]. Contamination by extraneous microbiota may be a challenge with culture-based approaches, rendering determination as to whether isolated microorganisms are pathogens or contaminants difficult based on identity alone (e.g., *S. epidermidis*) in some cases. This can be overcome by culturing multiple samples (e.g., periprosthetic tissues) from each patient; recovery of the same species from more than one sample typically implies that it is a cause of infection. Time to detection is another limitation of culture-based methods; for example, anaerobic bacterial cultures are routinely incubated for 14 days prior to being reported as negative [9,75]. A 2022 study of 536 PJI patients found that the median time-to-positivity for all positive cultures was almost 3.5 days, although this was dependent on the microbial species, with *S. aureus* growing in the shortest mean time (1 day) and *Cutibacterium acnes* in the longest mean time (almost 7 days). Sample type also impacted time to results, with synovial fluid having the shortest mean time-to-positivity, followed by periprosthetic soft tissues [75].

Molecular assays, such as nucleic acid amplification tests (typically, polymerase chain reaction (PCR) assays) and those based on microbial sequencing, are increasingly used for PJI diagnosis and pathogen identification [9,71,76,77,78,79]. As molecular techniques detect microbial components and not living bacteria, they may theoretically have higher sensitivities than culture-based assays. This increased sensitivity may come with drawbacks. Similar to culture-based assays, molecular techniques may be limited by low bacterial abundance and prior antimicrobial usage. Detection of contaminants or otherwise clinically insignificant bacterial components is a limitation of molecular-based techniques. Because of the sensitivity of these techniques, microbial elements from sample contamination during harvest and/or processing or left over from previous infections may be detected, leading to erroneous results. Molecular techniques may be more expensive and have longer turnaround times than culture-based techniques, although this is not always the case. Recently, sequencing-based approaches to microbial detection, based on targeted sequencing of the 16S ribosomal RNA gene and shotgun metagenomic sequencing, have been described for PJI diagnosis, with the former being more commonly used in current clinical practice [74,80,81,82,83]. In 2022, bioMérieux received US Food and Drug Administration (FDA) authorization for the BioFire^®^ Joint Infection Panel, which interrogates a single synovial fluid sample against a 31 microbial target panel in approximately one hour. A limitation of panel-based diagnostics is that it only detects microorganisms that are included in the panel; for example, the aforementioned panel does not include *S. epidermidis*, an important PJI pathogen [82]. Novel microbial-based detection techniques remain an area of interest for rapid PJI diagnosis and microbial identification.

### 2.2. Host-Based Diagnostic Techniques

Host-based biomarkers have been used to differentiate PJI from NIAF, beginning with the assessment of acute inflammation in periprosthetic tissue (a detailed review of which is beyond the scope of this manuscript). Elevated synovial fluid total nucleated cell count and polymorphonuclear (PMN) percentage were established as biomarkers for PJI early on (Table 1) [84,85,86,87,88,89,90,91,92,93,94,95,96,97,98,99,100,101,102,103,104,105,106,107,108,109,110,111,112,113]. Notably, time post-arthroplasty affects synovial fluid total nucleated cell count and PMN percentage; as such, the timing of sampling must be considered when interpreting results [114].

In 2019, the detection of α-defensin in synovial fluid by the lateral flow Synovasure™ PJI Test was approved by the FDA as the first FDA-approved host biomarker to aid in the detection of PJI [115,116]. α-defensin is an antimicrobial peptide primarily produced by neutrophils and macrophages and is thought to kill bacteria, fungi, and enveloped viruses by creating pores in microbial cell membranes [117,118]. The synovial fluid α-defensin lateral flow test generally exhibits good diagnostic accuracy for differentiating PJI from NIAF involving THA or TKA, with similar performance to an enzyme-linked immunosorbent assay (ELISA) [115], but performance may depend on which clinical definition for PJI is used, with lower correlation with EBJIS and IDSA, than MSIS definitions [119,120,121]. Diagnostic accuracy may be lower when analyzing arthroplasty types other than THA and TKA; for example, low sensitivity for PJI involving shoulder arthroplasties, which commonly involve *Cutibacterium acnes*, has been reported [122,123]. Use of synovial fluid α-defensin levels as sole indicators of PJI may be controversial [92,99,103,121,122,123,124,125,126,127,128,129]. A summary of α-defensin studies is included in Table 2 [96,98,99,102,103,104,106,115,120,121,124,126,130,131,132,133,134,135,136,137,138,139,140,141,142,143,144,145,146,147,148,149,150,151,152,153,154].

Beyond the assessment of acute inflammation in periprosthetic tissue, synovial fluid total nucleated cell count and PMN percentage, and synovial fluid α-defensin testing, other host biomarkers across various sample types may aid in the clinical determination of infection [99,155,156,157]. For example, synovial fluid C-reactive protein (CRP), calprotectin, interleukin (Il) 6 (Il-6), leukocyte esterase (LE), or lipocalin may be assessed. A summary of synovial fluid biomarkers is included in Table 3 [88,91,92,94,96,98,99,100,101,102,104,106,107,108,111,113,144,147,148,149,153,158,159,160,161,162,163,164,165,166,167,168,169,170,171,172,173,174,175,176,177,178,179,180,181,182,183,184,185]. Serum may also be evaluated by quantifying CRP, D-dimer, erythrocyte sedimentation rate (ESR), IL-6, or procalcitonin, which are often elevated in PJI [186,187,188,189,190]. With such single host biomarker assays, however, PJI diagnosis may be difficult in some cases, and information provided may be redundant between assays; further, underlying immune disorders, such as rheumatoid arthritis or other inflammatory diseases, and co-morbidities, may affect test performance [98,178,191,192,193,194,195]. While analysis of synovial fluid total nucleated cell count and polymorphonuclear percentage, and to a lesser extent, α-defensin, can be performed in most medical centers, synovial fluid testing for CRP, calprotectin, Il-6, LE, and lipocalin, may be more limited in availability.

Similar to PJI, there are no perfect assays for NIAF diagnosis. Mechanical-related failures, such as aseptic loosening and fractures, are typically diagnosed and distinguished by X-ray, though non-mechanical failures, such as instability and adverse tissue reaction, may be difficult to differentiate from PJI due to inflammatory responses at affected areas [11,12,59,60]. It has been suggested that some non-mechanical-related NIAF cases may actually represent infection [196,197,198], although recent work using culture, PCR, and deep microbial sequencing has shown that microorganisms are rarely found in NIAF [73,81,199,200,201]. Due to a lack of accurate NIAF diagnostic tools, a NIAF diagnosis may be contingent on the lack of a PJI diagnosis—that is, once arthroplasty failure is deemed to not be PJI-associated, NIAF is diagnosed. While helpful in determining whether antimicrobial treatment is necessary, the non-infectious pathogenesis underlying the failure may be unclear in such instances.

### 2.3. Importance of Fast and Accurate Arthroplasty Failure Diagnosis

The rise in arthroplasty procedures and the associated increase in PJI and NIAF, alongside the status of PJI diagnostics, justify the development of improved diagnostic techniques to differentiate PJI from NIAF and subsets within. Determining whether arthroplasty failure is due to PJI or NIAF and the causative organism, if infection is present, is important for selecting ideal management. As previously mentioned, results from currently used diagnostics may result in ambiguous classification, and in such cases, patients may receive unnecessarily broad-spectrum antimicrobial treatment or, alternatively, treatment that does not treat the actual cause of the arthroplasty failure. Unnecessary antimicrobial treatment may cause dysbiosis and drug-associated toxicity, in addition to aiding in the selection and expansion of antimicrobial-resistant bacteria, which is leading to dramatic effects on global health [53,202,203].

In 2022, the Antimicrobial Resistance Collaborators published an article estimating that 5 million deaths in 2019 worldwide were associated with bacterial antimicrobial resistance, although the full impact of antimicrobial resistance remains unknown due to the lack of global tracking systems. Methicillin-resistant *S. aureus*, an important cause of PJI, was a leading cause of death associated with antimicrobial resistance, resulting in more than 100,000 deaths in 2019 [204]. The World Health Organization (WHO) and Centers for Disease Control and Prevention (CDC) have included antimicrobial-resistant *S. aureus* on their Priority Pathogen and Urgent Threat lists, respectively, due to its impact on global health [205,206]. *S. epidermidis*, an important PJI pathogen, is associated with high rates of methicillin resistance. The CDC reported a greater than 15% increase in antimicrobial-resistant bacterial-associated infections and deaths in hospitals in 2020, possibly attributed to weakened infection prevention protocols and/or the usage of antibacterial agents during the COVID-19 pandemic [207].

These findings highlight the growing global crisis of antimicrobial resistance and illustrate the imperative of accurate and specific diagnoses of infectious diseases. Developing novel diagnostic techniques to differentiate PJI from NIAF and inform targeted antimicrobial usage will aid in patient management and in antimicrobial stewardship efforts, which will, in turn, assist in the fight against antimicrobial resistance globally.

## 3. Detailed Immune Response Profiling for Arthroplasty Failure Diagnosis

While evaluating individual host biomarkers may allow differentiation of PJI and NIAF in many instances, there remain cases that are clinically challenging to classify; expanded understanding as to how the full human immune system reacts during arthroplasty failure may provide insights into future diagnostic and possibly treatment opportunities. Immune profiling allows differentiation of PJI and NIAF, and may potentially identify subsets thereof, even in the context of inflammation related to surgical procedures or underlying inflammatory conditions. Recently, advances in multi-omics techniques have allowed a detailed characterization of the host immune response during PJI and NIAF (Figure 4).

### 3.1. Transcriptomic Immune Profiling

Transcriptomic analyses, such as RT-PCR and RNA-sequencing, have been conducted to interrogate the immune microenvironment during PJI and assess its potential impact on local bone and joint health. Transcriptomic studies performed on periprosthetic tissues from PJI show, unsurprisingly, that elevated expression of antimicrobial and immune cell activation genes dominates the immune response. In arthroplasty studies targeting specific transcripts, those associated with neutrophil activation, such as calprotectin, and IL-8, and macrophage inflammatory transcripts, such as chemokine (C-X-C motif) ligand (CXCL) 2 (CXCL2), and chemokine (C-C motif) ligand (CCL) 3 (CCL3), are elevated in PJI and associated with bone degeneration through bone-resorbing osteoclast generation, as well as induction of osteoblast inflammatory cytokine production [190,208,209]. Another targeted transcriptomic study of PJI-associated periprosthetic tissues found elevated levels of granulocyte colony-stimulating factor (G-CSF), IL-1β, IL-6, IL-8, and CD40L at infection sites [210]. In other targeted studies, levels of inflammatory mediators, such as toll-like receptor 2 (TLR-2), CCL2, presepsin, and osteopontin (OPN), were elevated in the serum of patients with PJI [186,211]. Of note, the last two are candidate biomarkers for sepsis diagnosis [212,213]. In addition to novel findings, many studies recapitulate antimicrobial-related inflammatory biomarkers already used in PJI diagnosis, such as α-defensin, IL-6, and D-dimer.

Untargeted transcriptomic analysis of sonicate fluid from PJI patients has also been conducted. In a 2022 study by Masters et al., sonicate fluid from hip and knee arthroplasty failures underwent RNA sequencing; overall differentially expressed gene (DEG) profiling separated PJI and NIAF samples by principal component analysis (PCA). Pathway analysis found elevated DEGs to be primarily related to host defense, immune response, and cellular development and repair of canonical pathways. In all, 28 previously described potential PJI biomarkers and three novel potential biomarkers, including CCL20, coagulation factor F7, and Fc receptor-like 4 (FCRL4), were elevated in PJI compared to NIAF [214].

### 3.2. Proteomic Immune Profiling

Proteomic analyses of PJI and NIAF samples have also been conducted to assess the functional output of the immune response during arthroplasty failure and investigate its potential diagnostic use. Proteomic, similar to transcriptomic, profiling of arthroplasty failure, found that the local immune response during PJI is primarily driven by elevated antimicrobial inflammatory proteins, while the proteome of NIAF samples is more related to tissue homeostasis and wound healing. For example, targeted proteomic immunoassays conducted on synovial fluid have found neutrophil elastase (ELA-2), bactericidal/permeability-increasing protein (BPI), lipocalin, lactotransferrin, thrombospondin, IL-1β, IL-10, IL-1α, lactate, interferon (IFN)γ, IL-5, and IL-17A to be elevated in PJI compared to NIAF [102,107,147,184]. The diagnostic accuracy of α-defensin, CRP, IL-6, and LE were unsurprisingly recapitulated. Expression of the antimicrobial complement cascade of proteins has also been studied in synovial fluid using multiplex immunoassay. Complement proteins C1q, C3b/C3i, C4b, C5, C5a, MBL, and properdin were elevated in the PJI compared to NIAF. Individually, C1q was most able to differentiate PJI from NIAF, although the combination of elevated C1q, C3b/C3i, C4b, C5, C5a, and MBL was most predictive of PJI [215].

Recently, we reported the characterization of the proteome of 200 sonicate fluid samples using a 92-target inflammatory protein panel not specifically designed for PJI [216]. Sixteen proteins were elevated in PJI, including CCL20, oncostatin M, extracellular newly identified receptor for advanced glycation end products binding protein (EN-RAGE), IL-6, IL-1α, IL-8, CXCL5, CXCL1, CXCL6, leukemia inhibitory factor (LIF), IL-17A, tumor necrosis factor (TNF), matrix metallopeptidase 1 (MMP-1), IFNγ, IL-18R1, and CCL4, and 21 proteins were elevated in NIAF, including macrophage-colony stimulating factor (CSF-1), osteoprotegerin, Flt3L, AXIN1, TNF-like weak inducer of apoptosis (TWEAK), TNF receptor superfamily member 9 (TNFRSF9), monocyte chemoattractant protein (MCP) 1 (MCP-1), complement C1r/C1s, Uegf, Bmp1 domain containing protein 1 (CDCP1), Skp, Cullin, F-box containing complex (SCF complex), eukaryotic translation initiation factor 4E (eIF4E)-binding protein 1 (4E-BP1), TNF-related activation-induced cytokine (TRANCE), CD40, MMP-10, sulfotransferase family 1A member 1 (ST1A1), MCP-4, IL-18, hepatocyte growth factor (HGF), IL-10RB, CCL3, signal transducing adaptor molecule binding protein binding protein (STAMBP), and CXCL10 [216]. While individual proteins were moderately to mildly predictive of PJI vs. NIAF (the most predictive being CCL20), a combination of elevated CCL20 and IL-8 and lowered MCP-1 and CCL3 was highly predictive. PCA differentially separated PJI and NIAF samples by overall proteomic profile. In addition to comparing all PJI to all NIAF samples, samples within PJI and NIAF subgroups were compared. Although proteomic profiling with the small panel studied was unable to detect differences between staphylococcal vs. non-staphylococcal PJI, or between aseptic loosening, instability, stiffness, osteolysis, or other causes of NIAF, two proteins were differentially expressed when comparing causative species of PJI, with elevated IL-17A in *S. aureus* compared to *S. epidermidis* and *Staphylococcus lugdunensis*-associated PJI, and elevated CCL11 in *S. epidermidis* compared to *S. aureus* and *Streptococcus agalactiae*-associated PJI [216]. These results, generated with a small protein panel, justify more extensive proteomic analyses of PJI with a view to determining whether more comprehensive proteomic profiles might be able to point to specific underlying potential PJI-causing pathogens.

To preliminarily characterize the proteome during PJI and NIAF in an untargeted manner, a subset of four *S. aureus*-associated PJI and four NIAF sonicate fluid samples that had undergone analysis using the 92-target inflammatory protein panel above [216] were analyzed using liquid chromatography with tandem mass spectrometry (LC-MS/MS) [217]. Of 810 proteins quantified, 35 were differentially abundant in *S. aureus* PJI and NIAF samples. PCA differentially separated the overall proteomic profiles of *S. aureus* PJI and NIAF sonicate fluid samples. Gene ontology pathway analysis found *S. aureus* PJI to be associated with elevated proteins in microbial defense response pathways, specifically those related to neutrophil degranulation and activation. Proteins within molecular function pathways related to endopeptidase and peptidase activity, transition metal and iron ion binding, and TLR and receptor for advanced glycation endproducts (RAGE) receptor binding were also elevated in PJI compared to NIAF. In all, fifteen proteins were elevated in PJI, including lactotransferrin, lipocalin, myeloperoxidase, calprotectin A8 and A9 subunits, cathepsin G, neutrophil elastase (ELA-2), eosinophil cationic protein (RNASE3), endoplasmic reticulum to nucleus signaling 1 (ERN1), matrix metalloproteinase-9 (MMP-9), lysozyme C, haptoglobin, lamin-B1, glycogen phosphorylase, liver form (PYGL), leucine-rich α-2-glycoprotein (LRG1). Twenty proteins were elevated in NIAF, including cartilage acidic protein 1 (CRTAC1), melanoma cell adhesion molecule (MCAM), IFI30 lysosomal thiol reductase (IFI30), osteopontin, β-hexosaminidase subunit β (HEXB), proteoglycan 4, pancreatic ribonuclease (RNASE1), dermcidin, CD44, annexin A2, serpin B6, branched-chain-amino-acid aminotransferase (BCAT1), dihydrolipoamide S-succinyltransferase (DLST), shock protein β-1 (HSPB1), early endosome antigen 1 (EEA1), collagen α-2(I) chain (COL1A2), fatty acid-binding protein, epidermal (FABP5), fructose-1,6-bisphosphatase 1 (FBP1), fatty acid-binding protein, heart (FABP3), and cathepsin D [217].

### 3.3. Cellular Immune Profiling

While the elevation of leukocytes, particularly neutrophils, is well-established in PJI, a robust understanding of the cellular profile during PJI is still being investigated. Cellularity profiling has primarily been conducted using synovial fluid and periprosthetic tissue. Due to the effects of processing, direct cellularity studies are limited when using sonicate fluid. To circumvent this, the transcriptomic results from the previously described Masters et al., *2022* bulk RNA-sequencing study on sonicate fluid were subjected to bioinformatic cellular deconvolution using CIBERSORTx [214,218]. Cellular deconvolution allows cellular analysis by “unmixing” bulk transcriptomic data to generate predicted cellularity profiles, in this case, made of 22 specific cell-types. Cellularity profiles created by CIBERSORTx are differentially clustered by PCA between PJI and NIAF. The differentiation of PJI and NIAF by predicted cellularity profiling was mainly separated by roles during inflammation—that is, cell types important for antimicrobial immunity were elevated in PJI, while NIAF populations were primarily composed of immune cells involved in tissue homeostasis and repair. In all, predicted total granulocyte, neutrophils, activated mast cells, CD8+ T cells, eosinophils, resting NK cells, activated CD4+ memory T cells were elevated in PJI, with predicted total macrophages/monocytes, M0 macrophages, M2 macrophages, total B cells, plasma cells, regulatory T cells, naïve B cells, and follicular helper T cells elevated in NIAF. Total granulocytes, neutrophils, and activated mast cells were most predictive of PJI from NIAF [218].

While it is known that infiltrating neutrophils are elevated during PJI, the role of mast cells during arthroplasty failure is uncharacterized. Tissue-resident mast cells have been described as “sentinel cells” able to detect microbial insults and initiate downstream antimicrobial inflammation by recruiting neutrophils and presenting bacterial antigens to the adaptive immune response, in additional to killing bacteria through secretion of antimicrobial peptides [219,220]. Activation of joint-specific mast cells has been linked to the induction of rheumatoid arthritis and increased joint inflammation during osteoarthritis and arthrofibrosis [174,221,222,223]. As such, it could be reasoned that mast cells may play a role in antimicrobial host defense during PJI. Further investigation of the possible presence of this cell type in PJI is needed.

Findings from cellular deconvolution analysis of sonicate fluid have been largely recapitulated by results of flow cytometry experiments on synovial fluid and periprosthetic tissue. Anti-bacterial granulocytes, primarily driven by neutrophils and eosinophils, NK cells, and monocytes, were elevated in PJI vs. NIAF synovial fluid samples [224]. Similar to findings in synovial fluid, macrophages and monocytes were elevated in PJI-associated compared to NIAF periprosthetic tissues.

Although not normally characterized as an immune cell type, elevated platelets have been reported in the blood of patients with PJI; their diagnostic usefulness remains controversial [225,226,227]. There have been conflicting reports regarding the presence of T cells in arthroplasty failure, with findings ranging from lower or no to increased T cells in PJI compared to NIAF [210,224]. Whether, and if, T cells play a role during arthroplasty failure warrants future investigation. It has been suggested that the inflammatory response to bacterial biofilms during PJI may lead the recruitment of anti-inflammatory myeloid-derived suppressor cells (MDSCs) to the joint, leading to immune response suppression and downstream chronic infection. Recruitment of MDSCs to the joint is elevated during PJI, likely due to the production of Il-12 by the host or of lactate and ATP synthase by certain biofilms themselves. Recruitment of MDSCs leads to the suppression of antimicrobial phagocyte recruitment and inhibits bacterial clearance [210,224,228,229,230]. The role of MDSCs in arthroplasty failure is an area for future research.

### 3.4. Limitations of Immune Profiling for Arthroplasty Failure Diagnosis

In addition to the limitations of each individual technique to profile immune response during arthroplasty failure, there are limits associated with detailed immune profiling and its overall potential diagnostic use. At this time, virtually all studies profiling immune response during arthroplasty failure have been conducted as research studies. Though some results may portend future clinical use, when and whether these techniques can be validated for clinical use is unknown. Cohort sizes have been relatively small, with limited comorbidities addressed. As such, whether currently identified immune profiles will be recapitulated in larger, more diverse clinical populations is unknown. These studies have typically been conducted on samples from patients with clear PJI or NIAF diagnoses. It is unknown how these approaches will perform in more challenging to diagnose cases, the very cases where improved diagnostics are needed. The logistics of conducting multi-omics analyses may also be problematic in clinical scenarios. Currently, these techniques and the necessary bioinformatic analyses are expensive and time-consuming. Prices of multi-omics analyses may decrease over time; the introduction of individualized medical tools, including personalized computational diagnostics, may become commonplace in the future.

A selection of cofactors—most not-yet-investigated—that may be relevant to accurate and reproducible results when assessing the immune response during arthroplasty failure is shown in Table 4. These variables may be important to note when planning, conducting, and analyzing the results of immune response profiling studies. Ways in which such variables impact immune responses to arthroplasty failure remain largely unexplored. As advanced multi-omic analyses develop, it will be interesting to investigate the clinical and diagnostic impact of these variables and their interactions.

## 4. The Future of PJI and NIAF Diagnostics

Multi-omics techniques to characterize immune response during arthroplasty failure represent a novel approach to potential future diagnosis of PJI and NIAF. While approaches described here differentiate PJI from NIAF, none have been discriminatory enough to define the underlying infectious organism(s) or cause of non-infectious failure, topics that deserve further study. Host-based diagnostics are not necessarily replacements for microbial-based detection tools, but will likely, instead, complement them. Complex bioinformatic tools, in combination with computational techniques, such as machine learning and artificial intelligence, are at the cutting edge of individualized diagnostics [231,232,233,234]. Advanced computational studies have already been conducted to better understand their potential use in PJI prediction [235,236,237,238]. These tools may be useful for addressing the current limitations of host-based profiling and synthesizing descriptive diagnostic readouts from multi-omics results in the future. Simultaneous assessment of host response and microorganism detection may allow single diagnostic reports to determine causative pathogen(s) and assess biological responses, providing insight into underlying key inflammatory etiologies and informing precision treatment (Figure 5). Continued development of tools to detect causes of arthroplasty failure remains a challenge that warrants ongoing collaborative investigation.

## Figures and Tables

**Figure 1 antibiotics-12-00296-f001:**
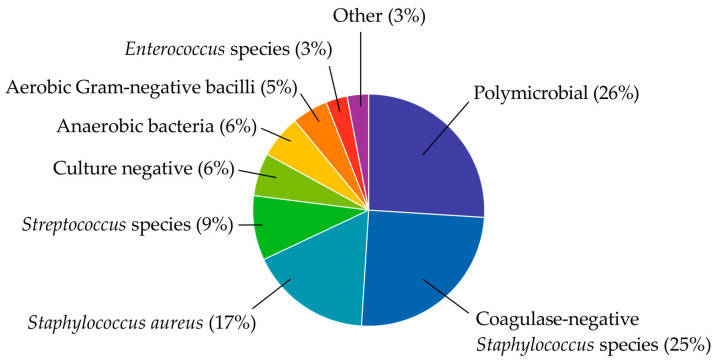
Causes of periprosthetic joint infection after total hip and total knee arthroplasty based on data from Tai, D.B.G. et al. *Clin Microbiol Infect*
**2022**, *28*, 255–259 [50]. Graph created in GraphPad Prism v9.4.0 (San Diego, CA, USA).

**Figure 2 antibiotics-12-00296-f002:**
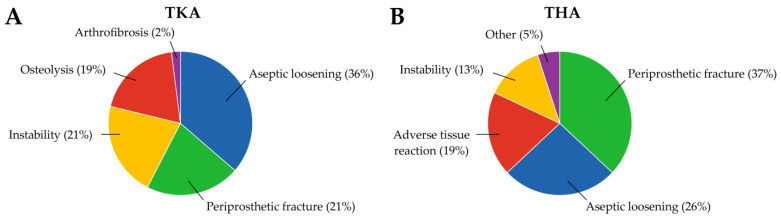
Common causes of non-infectious arthroplasty involving (**A**) total knee arthroplasty (TKA) and (**B**) total hip arthroplasty (THA). TKA data are from Abdel, M.P. et al. *Bone Joint J*
**2017**, 99-B, 647–652 [64]. THA data from Ledford, C.K. et al. *J Am Acad Orthop Surg*
**2019**, 27 [12]. Graph created in GraphPad Prism v9.4.0 (San Diego, CA, USA).

**Figure 3 antibiotics-12-00296-f003:**
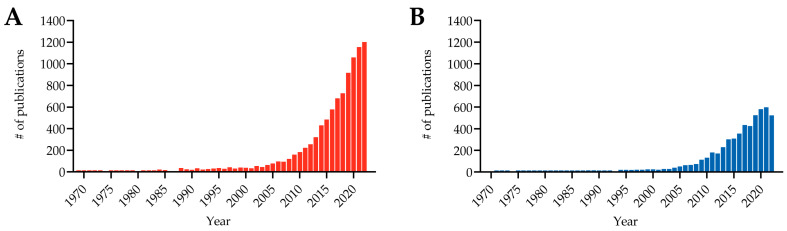
Periprosthetic joint infection (PJI)-related publication counts between 1969 and 2022. (**A**) All PJI-related publications were calculated based on a PubMed query with keywords “PJI”, “prosthetic joint infection”, or “periprosthetic joint infection”. (**B**) PJI diagnosis-related publications were calculated based on a PubMed query with keywords “PJI diagnosis”, “prosthetic joint infection diagnosis”, or “periprosthetic joint infection diagnosis”. Query conducted on 16 January 2023. Graph created in GraphPad Prism v9.4.0 (San Diego, CA, USA).

**Figure 4 antibiotics-12-00296-f004:**
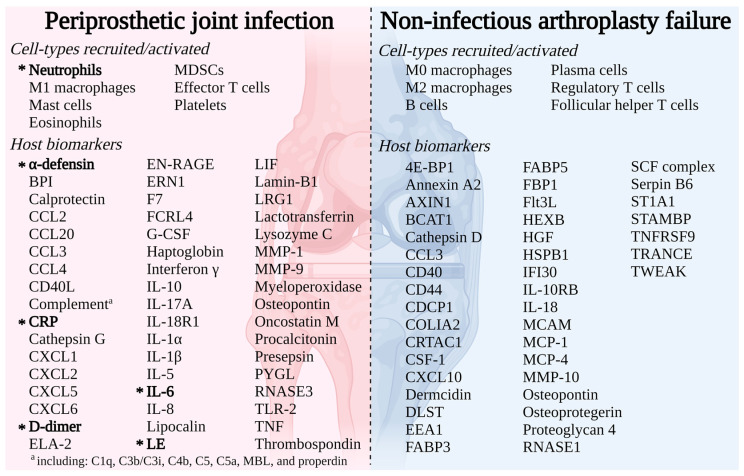
Immune response to arthroplasty failure due to periprosthetic joint infection (PJI) or non-infectious arthroplasty failure (NIAF) detailing cell-types recruited/activated and host markers. Bolded, starred (*) entries represent those currently used for the diagnosis of PJI. Abbreviations used: 4E-BP1, eukaryotic translation initiation factor 4E (eIF4E)-binding protein 1; BCAT1, branched-chain-amino-acid aminotransferase; BPI, bactericidal/permeability-increasing protein; CCL, chemokine (C-C motif) ligand; CDCP1, complement C1r/C1s, Uegf, Bmp1 domain containing protein 1; COL1A2, collagen α-2(I) chain; CRTAC1, cartilage acidic protein 1; CRP, C-reactive protein; CSF-1, macrophage-colony stimulating factor; CXCL, chemokine (C-X-C motif) ligand; DLST, dihydrolipoamide S-succinyltransferase; EEA1, early endosome antigen 1; ELA-2, neutrophil elastase; EN-RAGE, extracellular newly identified receptor for advanced glycation end products binding protein; ERN1, endoplasmic reticulum to nucleus signaling 1; FABP3, fatty acid-binding protein–heart; FABP5, fatty acid-binding protein–epidermal; FBP1, fructose-1,6-bisphosphatase 1; FCRL4, Fc receptor-like 4; G-CSF, granulocyte colony-stimulating factor, HEXB, β-hexosaminidase subunit β; HGF, hepatocyte growth factor; HSPB1, heat shock protein β-1; IFI30, IFI30 lysosomal thiol reductase; IL, interleukin; LE, leukocyte esterase; LIF, leukemia inhibitory factor; LRG1, leucine-rich α-2-glycoprotein; MBL, mannose-binding lectin; MCAM, melanoma cell adhesion molecule; MCP, monocyte chemoattractant protein; MDSC, myeloid derived suppressor cell; MMP, matrix metallopeptidase; PYGL, glycogen phosphorylase–liver; RNASE1, pancreatic ribonuclease; RNASE3, eosinophil cationic protein; SCF complex, Skp, Cullin, F-box containing complex; TLR-2, toll-like receptor 2; TNF, tumor necrosis factor; TNFRSF9, TNF receptor superfamily member 9; TRANCE, TNF-related activation-induced cytokine; TWEAK, TNF-like weak inducer of apoptosis; ST1A1, sulfotransferase family 1A member 1; STAMBP, signal transducing adaptor molecule binding protein. Created with Biorender.com.

**Figure 5 antibiotics-12-00296-f005:**
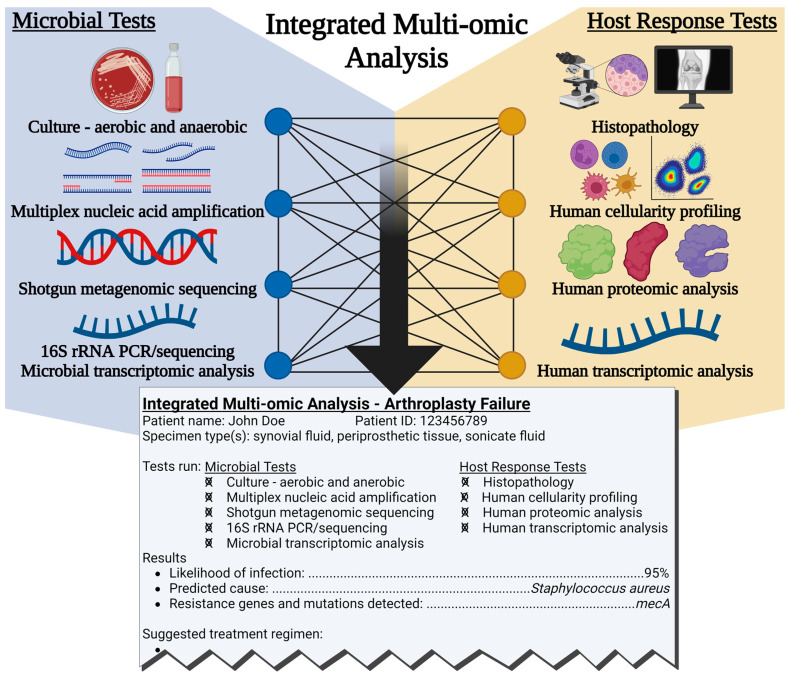
Potential multi-omics diagnostic scheme using simultaneous microorganism detection and assessment of host response to determine whether infection is present, and if so, to define the causative pathogen(s) and inform treatment. Created with Biorender.com.

**Table 1 antibiotics-12-00296-t001:** Sensitivity and specificity of synovial fluid total nucleated cell count and polymorphonuclear percentage for periprosthetic joint infection diagnosis.

Biomarker	Knee/Hip/Other	Time Since Arthroplasty	Cut-Point	Sensitivity (%) ^a^	Specificity (%) ^a^	Citation
Total nucleated cell count—cutoff values in cells/µL						
Mason et al., 2003	440/-/-	NR	2500 50,000	69 19	98 100	[84]
Trampuz et al., 2004	133/-/-	>6 months	1700	94 (80–99)	88 (80–93)	[85]
Zmistowski et al., 2012	153/-/-	NR	3000	94	93	[86]
Dinneen et al., 2013	48/27/-	NR	1590	90 (78–100)	91 (83–100)	[87]
Wyles et al., 2013	-/39/-	NR	3000	100 (40–100)	57 (85–100)	[88]
Gallo et al., 2017	203/188/-	>7 months	3450	95	95	[89]
Higuera et al., 2017	-/453/-	≥3 months	3966	90	91	[90]
Kim et al., 2017	197/-/-	>7 days	11,200 16,000	100 (73–100) 75 (43–95)	99 (96–100) 100 (98–100)	[91]
Lee et al., 2017	33 studies	Pooled	Pooled	89 (86–91)	86 (80–90)	[92]
Shahi et al., 2017	836 total	NR	10,000	86	83	[93]
Sousa et al., 2017	40/15/-	>1 month	1463 2064	100 91	7275	[94]
Balato et al., 2018	250/-/-	>90 days	3000	81 (74–86)	91 (86–95)	[95]
De Vecchi et al., 2018	45/21/-	NR	1600 3000	100 (87–100) 94 (78–99)	82 (65–93) 91 (75–98)	[96]
Kuo et al., 2018	131/83/-	NR	835	84 (65–96)	78 (72–84)	[97]
Tahta et al., 2018	38/-/-	>3 months	2347	86 (70–100)	76 (63–98)	[98]
Carli et al., 2019	26 studies	Pooled	Pooled	93	90	[99]
Dijkman et al., 2020	80/-/-	NR	2575	92	84	[100]
Mihalič et al., 2020	25/24/-	NR	1700	82 (55–100)	97 (92–100)	[101]
Sharma et al., 2020	93/14/-	NR	1100	89	98	[102]
Ivy et al., 2021	74/25/-	NR	1700	83 (59–96)	81 (70–89)	[103]
Levent et al., 2021	143/116/-	NR	3000	88	88	[104]
van den Kieboom et al., 2021	43/101/-	NR	3000 4552	87 (66–97) 86	78 (66–87) 85	[105]
Baker et al., 2022	358/36/-	>90 days	3000	92	99	[106]
Huang et al., 2022	39/39/-	NR	3005	90 (78–97)	100 (88–100)	[107]
Lazic et al., 2022	4/10/-	NR	4550	40 (12–74)	100 (79–100)	[108]
Dilley et al., 2023	485/245/-	>6 weeks	5600	72	86	[109]
Polymorphonuclear (PMN) percentage—cutoff values in % of total white blood cell count						
Mason et al., 2003	440/-/-	NR	6080	7657	89100	[84]
Trampuz et al., 2004	133/-/-	>6 months	65	97 (85–100)	98 (93–100)	[85]
Zmistowski et al., 2012	153/-/-	NR	75	83	88	[86]
Dinneen et al., 2013	48/27/-	NR	65	90 (80–100)	87 (76–97)	[87]
Wyles et al., 2013	-/39/-	NR	80	100 (40–100)	97 (81–100)	[88]
Gallo et al., 2017	203/188/-	>7 months	75	93	91	[89]
Higuera et al., 2017	-/453/-	≥3 months	80	92	86	[90]
Lee et al., 2017	33 studies	Pooled	Pooled	89 (82–93)	86 (77–92)	[92]
Sousa et al., 2017	40/15/-	>1 month	7881	8778	7275	[94]
Balato et al., 2018	250/-/-	>90 days	80	84 (77–89)	95 (90–98)	[95]
Mihalič et al., 2020	25/24/-	NR	65	82 (55–100)	97 (92–100)	[101]
Qin et al., 2020	24/26/-	NR	70	92 (74–99)	80 (59–93)	[110]
Qin et al., 2020	45/48/-	>6 weeks	70	89 (75–97)	84 (72–92)	[111]
Sharma et al., 2020	93/14/-	NR	72	92	91	[102]
Ivy et al., 2021	74/25/-	NR	65	90 (65–99)	87 (78–94)	[103]
van den Kieboom et al., 2021	43/101/-	NR	80	79	63	[105]
Wang et al., 2021	45/48/-	>6 weeks	70	95 (82–99)	93 (83–98)	[112]
Qin et al., 2022	30/40/-	>2.5 years	70	89	80	[113]
Dilley et al., 2023	485/245/-	>6 weeks	82	81	78	[109]

^a^ % and 95% confidence interval, if reported; NR, not reported.

**Table 2 antibiotics-12-00296-t002:** Sensitivity and specificity of synovial fluid α-defensin for periprosthetic joint infection diagnosis.

Assay	Knee/Hip/Other	Cut-Point	Sensitivity (%) ^a^	Specificity (%) ^a^	Citation
Lateral flow					
Bingham et al., 2014	61/-/-	NA	100 (79–100)	95 (83–99)	[130]
Kasparek et al., 2016	29/11/-	NA	67 (35–89)	93 (75–99)	[131]
Sigmund et al., 2017	17/30/-	NA	69 (46–92)	94 (86–100)	[132]
Okroj et al., 2018	-/26/-	NA	100	68	[133]
Berger et al., 2017	85/36/-	NA	97 (85–100)	97 (90–99)	[134]
Suda et al., 2017	19/11/-	NA	77	82	[135]
Balato et al., 2018	51/-/-	NA	88 (75–95)	97 (87–100)	[136]
de Saint Vincent et al., 2018	23/13/3	NA	89	91	[137]
Gehrke et al., 2018	99/96/-	NA	92 (84–97)	100 (97–100)	[126]
Renz et al., 2018	151/61/-	NA	84 (71–94)	96 (92–99)	[120]
Riccio et al., 2018	49/22/2	NA	85 (70–94)	97 (84–100)	[138]
Sigmund et al., 2018	54/17/-	NA	77 (49–92)	98 (90–100)	[121]
Stone et al., 2018	121/62/-	NA	81 (65–92)	96 (91–99)	[139]
Tahta et al., 2018	38/-/-	NA	92 (80–100)	98 (90–100)	[98]
Plate et al., 2018	60/49/-	NA	90 (68–99)	92 (85–97)	[140]
Carli et al., 2019	9 studies	NA	96	82	[99]
Sigmund et al., 2019	48/53/-	NA	69 (51–83)	94 (85–98)	[141]
Sharma et al., 2020	93/14/-	NA	88	95	[102]
Abdo et al., 2021	53/-/-	NA	86 (65–97)	100 (89–100)	[142]
de Saint Vincent et al., 2021	59/39/8	NA	96	91	[143]
Deirmengian et al., 2021	203/102/-	NA	94 (84–99)	95 (91–97)	[115]
Ivy et al., 2021	74/25/-	NA	83 (59–96)	94 (86–98)	[103]
Yu et al., 2021	82/48/-	NA	83	86	[144]
Zeng et al., 2021	1443 total (pooled)	NA	83 (77–88)	95 (93–97)	[145]
Baker et al., 2022	358/36/-	NA	99	87	[106]
Kuiper et al., 2022	-/57/-	NA	83 (36–100)	92 (81–98)	[146]
Enzyme-linked immunoassay (ELISA)—cutoff values in mg/L					
Deirmengian et al., 2014	84/11/-	4.8	100 (88–100)	100 (95–100)	[147]
Deirmengian et al., 2014	116/33/-	5.2	97 (86–100)	96 (90–99)	[148]
Deirmengian et al., 2015	43/3/-	1.6	100 (85–100)	100 (85–100)	[149]
Frangiamore et al., 2016	78 total(1st stage) 38 total (2nd stage)	5.2 5.2	100 (86–100) 67 (12–95)	98 (90–100) 97 (83–99)	[150]
Bonanzinga et al., 2017	65/91/-	5.2	97 (92–99)	97 (92–99)	[151]
De Vecchi et al., 2018	45/21/-	5.2	84 (67–94)	94 (79–99)	[96]
Sigmund et al., 2018	54/17/-	5.2	85 (56–97)	98 (90–100)	[121]
Carli et al., 2019	9 studies	Pooled	97	87	[99]
Kleiss et al., 2019	112/90/-	5.2	78 (67–89)	97 (93–99)	[152]
Abdo et al., 2021	53/-/-	5.2	96 (77–100)	100 (89–100)	[142]
Deirmengian et al., 2021	203/102/-	5.2	89 (76–96)	98 (94–99)	[115]
Ivy et al., 2021	74/25/-	5.2	83 (59–96)	96 (90–99)	[103]
Levent et al., 2021	143/116/-	5.2	92	92	[104]
Li et al., 2021	17/33	35.5	96	100	[153]
Mass spectrometry					
Iorio et al., 2021	88/50/-	5.2 mg/L	93 (85–98)	96 (89–99)	[154]
Balato et al., 2022	125/-/-	1 µg/L	100 (96–100)	97 (90–98)	[124]

^a^ % and 95% confidence interval, if reported; NA, not applicable.

**Table 3 antibiotics-12-00296-t003:** Sensitivity and specificity of synovial fluid C-reactive protein (CRP), calprotectin, interleukin-6 (Il-6), leukocyte esterase (LE), and lipocalin for periprosthetic joint infection diagnosis.

Biomarker	Knee/Hip/Other	Cut-Point	Sensitivity (%) ^a^	Specificity (%) ^a^	Citation
C-reactive protein (CRP)—cutoff values in mg/L					
Parvizi et al., 2012	43/12/-	9.5	83	95	[158]
Parvizi et al., 2012	66/-/-	3.7	84	97	[159]
Wyles et al., 2013	-/39/-	8	75 (19–99)	68 (50–83)	[88]
Deirmengian et al., 2014	84/11/-	12.2	90 (73–98)	97 (90–100)	[147]
Deirmengian et al., 2014	116/33/-	3	98 (86–100)	79 (70–86)	[148]
De Vecchi et al., 2016	84/45/-	10	82 (61–93)	94 (87–98)	[160]
Kim et al., 2017	197/-/-	34.9 74.5	100 (74–100) 58 (28–85)	91 (83–95) 100 (97–100)	[91]
Lee et al., 2017	33 studies	Pooled	85 (78–90)	88 (78–94)	[92]
Sousa et al., 2017	40/15/-	1.6 6.7 8.0	91 78 74	88 94 97	[94]
De Vecchi et al., 2018	45/21/-	1.0	88 (70–96)	97 (83–100)	[96]
Gallo et al., 2018	116/124/-	8.8	92 (73–99)	100 (95–100)	[161]
Tahta et al., 2018	38/-/-	11.7	76 (62–97)	90 (80–100)	[98]
Carli et al., 2019	9 studies	Pooled	93	89	[99]
Plate et al., 2019	91/80/21	2.9	88	82	[162]
Sharma et al., 2020	93/14/-	5.6	80	92	[102]
Baker et al., 2022	358/36/-	6.9	74	98	[106]
Grzelecki et al., 2021	50/35/-	6.9	64	95	[163]
Li et al., 2021	17/33/-	9.0	76	96	[153]
Wang et al., 2021	36/61/-	7.3	85 (70–94)	93 (83–98)	[164]
Praz et al., 2021	91/102/-	2.74.4	85 (71–93) 83 (71–94.3)	77 (68–84) 88 (82–94)	[165]
Qin et al., 2022	30/40/-	11.6	89	49	[113]
Calprotectin—cutoff values in mg/L					
Wouthuyzen-Bakker et al., 2017	10/45/6	50 (LF)	89 (69–98)	90 (78–96)	[166]
Wouthuyzen-Bakker et al., 2018	12/21/1	50 (LF)	87 (60–98)	92 (78–98)	[167]
Salari et al., 2020	76/-/-	50 ELISA	100 (100–100)	95 (89–100)	[168]
Trotter et al., 2020	17/42/-	14 (LF)	75 (53–90)	76 (60–87)	[169]
Grzelecki et al., 2021	50/35/-	1.5	96	95	[163]
Warren et al., 2021, 2022	123/-/-	14 (LF) 14 (ELISA) 50 (LF) 50 (ELISA)	98 98 98 98	87 83 96 96	[170,171]
Cheok et al., 2022	5 studies	Pooled	94 (82–98)	93 (85–97)	[172]
Grassi et al., 2022	93/-/-	50 (LF) 50 (ELISA)	97 (87–100) 92 (79–98)	94 (84–99) 100 (93–100)	[173]
Hantouly et al., 2022	8 studies	Pooled	92 (84–98)	93 (84–99)	[174]
Lazic et al., 2022	4/10/-	50 (LF)	67 (40–93)	79 (57–100)	[108]
Xing et al., 2022	7 studies	Pooled	94 (87–98)	93 (87–96)	[175]
Interleukin-6 (Il-6)—cutoff values in ng/mL					
Deirmengian et al., 2014	84/11/-	2.3	89 (71–98)	97 (89–100)	[147]
Lee et al., 2017	33 studies	Pooled	81 (70–89)	94 (88–97)	[92]
Xie et al., 2017	8 studies	Pooled	91 (82–96)	90 (84–95)	[176]
Gallo et al., 2018	116/124/-	21.0	68 (47–85)	95 (87–99)	[161]
Carli et al., 2019	5 studies	Pooled	97	84	[99]
Mihalič et al., 2020	25/24/-	2.3	73 (45–100)	95 (87–100)	[101]
Qin et al., 2020	45/48/-	1.86	95 (82–99)	93 (83–98)	[111]
Sharma et al., 2020	93/14/-	0.417	74	88	[102]
Cheok et al., 2022	6 studies	Pooled	86 (74–92)	94 (90–96)	[172]
Li et al., 2022	30 studies	Pooled	87 (75–93)	90 (85–93)	[177]
Qin et al., 2022	63/39/-	1.3	90 (74–97)	89 (73–96)	[178]
Qin et al., 2022	30/40/-	2.0	91	97	[113]
Su et al., 2022	78/102/-	1.2	91 (79–97)	52 (38–66)	[179]
Leukocyte esterase (LE)					
Deirmengian et al., 2015	43/3/-	+	69 (41–89)	100 (84–100)	[149]
De Vecchi et al., 2016	84/45/-	+	93 (74–99)	97 (91–99)	[160]
Lee et al., 2017	33 studies	Pooled	77	95	[92]
Shahi et al., 2017	659 total	+	75	91	[180]
De Vecchi et al., 2018	45/21/-	+ + +	94 (79–99) 56 (38–56)	97 (83–100) 100 (87–100)	[96]
Wang et al., 2018	11 studies	Pooled	90 (76–96)	97 (95–98)	[181]
Carli et al., 2019	9 studies 10 studies	+ + +	97 84	9396	[99]
Dijkman et al., 2020	89/-/-	+ +	39	88	[100]
Sharma et al., 2020	93/14/-	+ +	81 90	9584	[102]
Chisari et al., 2021	226/33/-	+ + +	74 51	91100	[182]
Grzelecki et al., 2021	50/35/-	+ +	82	98	[163]
Levent et al., 2021	143/116/-	+ +	78	91	[104]
Yu et al., 2021	82/48/-	+ +	80	95	[144]
Grassi et al., 2022	93/-/-	+	46 (30 -63)	94 (84–99)	[173]
Logoluso et al., 2022	21/58/-	+	82	99	[183]
Lipocalin					
Vergara et al., 2018	54/18/-	152 ng/mL	86	77	[184]
Dijkman et al., 2020	89/-/-	740 ng/mL	92	83	[100]
Li et al., 2021	17/33/-	763 ng/mL	100	100	[153]
Huang et al., 2022	39/39/-	263 ng/mL	93 (77–99)	98 (89–100)	[107]
Svoboda et al., 2022	56/33/-	998 µg/mL	100	100	[185]

^a^ % and 95% confidence interval, if reported; LF, lateral flow test; ELISA, enzyme-linked immunoassay.

**Table 4 antibiotics-12-00296-t004:** Variables to consider when profiling the immune response to arthroplasty failure.

Patient-Related	Sample-Related	Treatment-Related	Failure-Related
Age Sex Time post-surgery Co-morbidities Implant site Initial reason for arthroplasty	Analysis method Specimen type Specimen processing Specimen age Specimen storage conditions	Prior antimicrobial treatment Antimicrobial agent type Treatment duration Primary or revision arthroplasty	Infectious or non-infectious Causative species/strain (PJI) Duration of infection (PJI) Mechanical ^a^ or non-mechanical ^b^ failure (NIAF)

^a^ Aseptic loosening, periprosthetic fracture; ^b^ Instability, adverse tissue reaction; PJI, periprosthetic joint infection; NIAF, non-infectious arthroplasty failure.

## Data Availability

Not applicable.

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
