# Peer review of "Profiling the Immune Response to Periprosthetic Joint Infection and Non-Infectious Arthroplasty Failure"

_antibiotics, 2023, doi:10.3390/antibiotics12020296_

Round 1

Reviewer 1 Report

The present review is a presentation of the current state of knowledge in linking the immune system with septic and aseptic arthroplasty failure.

Overall, it is a valuable piece of work that requires only minor rework.

The following adjustments should be made:

1.       The figures are listed in the text before the citation in each case. This is a hindrance to fluent reading and reduces comprehensibility.

2.       Figure 3 shows the chronological increase in publications on the topic of diagnosis of PJI. Like almost every field of research in medicine, there has been a drastic increase. A statement about the subject area: diagnosis of PJI cannot be made without a comparison to, for example, the main subject area of PJI and leads the reader to erroneous conclusions. Here the graph should be put in relation or not listed at all.

3.       Table 2 is unnecessarily interrupted in the current version, this should be considered in the final version.

4.       The last table is declared as table 3 whereas it should be table 4. Furthermore, it is a pure enumeration of variables without putting them into a clinical context. Here it would be very interesting for clinically active readers how the variables are to be considered and/or which modulatory effect on the immune response is to be expected (if then currently known).

Reviewer 2 Report

This is a well written, comprehensive report, shedding light on a severe difficulty in clinical management of artificial joint failures. It is very long and hard to read, but can serve as a reference for learners of orthopedics or infectious diseases. Many of the methods are not widely available, and I suggest a 2 or 3 column table, separating the investigations that are available in most medical centers where joint replacements are performed, from ones which are only available in sophisticated tertiary centers, from the ones that are not in use yet.

Figure 2: Pie charts. Colors should correspond with the issue rather than with its frequency.

Table 1 & 2: hip, knee and both should be stated as + or - in separate columns, while the number of the reference an be put in brackets with the author's name in column 1. Number of cells per cubic micrometer can be mentioned once, in the heading of the column.

Table 4 and not 3!!!
